# Patient-derived iPSCs link elevated mitochondrial respiratory complex I function to osteosarcoma in Rothmund-Thomson syndrome

**Brittany E. Jewell**[1,2], **An Xu**[1], **Dandan Zhu**[1], **Mo-Fan Huang**[1,2], **Linchao Lu**[3], **Mo Liu**[1], **Erica L. Underwood**[4], **Jun Hyoung Park**[5], **Huihui Fan**[6], **Julian A. Gingold**[7], **Ruoji Zhou**[1], **Jian Tu**[1], **Zijun Huo**[1], **Ying Liu**[1], **Weidong Jin**[3], **Yi-Hung Chen**[8], **Yitian Xu**[9], **Shu-Hsia Chen**[9], **Nino Rainusso**[3], **Nathaniel K. Berg**[2,10], **Danielle A. Bazer**[11], **Christopher Vellano**[12], **Philip Jones**[12], **Holger K. Eltzschig**[2,10], **Zhongming Zhao**[2,6], **Benny Abraham Kaipparettu**[5], **Ruiying Zhao**[1], **Lisa L. Wang**[3☯]*, **Dung-Fang Lee**[1,2,6,13☯]*

**1** Department of Integrative Biology and Pharmacology, McGovern Medical School, The University of Texas Health Science Center at Houston, Houston, Texas, United States of America, **2** The University of Texas MD Anderson Cancer Center UTHealth Graduate School of Biomedical Sciences, Houston, Texas, United States of America, **3** Department of Pediatrics, Baylor College of Medicine, Texas Children's Hospital, Houston, Texas, United States of America, **4** Department of Neurobiology and Anatomy, McGovern Medical School, The University of Texas Health Science Center at Houston, Houston, Texas, United States of America, **5** Department of Molecular and Human Genetics, Baylor College of Medicine, Houston, Texas, United States of America, **6** Center for Precision Health, School of Biomedical Informatics, The University of Texas Health Science Center at Houston, Houston, Texas, United States of America, **7** Department of Obstetrics & Gynecology and Women's Health, Einstein/Montefiore Medical Center, New York City, New York, United States of America, **8** Department and Institute of Pharmacology, National Yang Ming Chiao Tung University, Taipei, Taiwan, **9** Center for Immunotherapy Research, Cancer Center of Excellence, Houston Methodist Research Institute, Houston, Texas, United States of America, **10** Department of Anesthesiology, McGovern Medical School, The University of Texas Health Science Center at Houston, Houston, Texas, United States of America, **11** Department of Neurology, Renaissance School of Medicine at Stony Brook University, Stony Brook, New York, United States of America, **12** TRACTION Platform, Therapeutics Discovery Division, University of Texas MD Anderson Cancer Center, Houston, Texas, United States of America, **13** Center for Stem Cell and Regenerative Medicine, The Brown Foundation Institute of Molecular Medicine for the Prevention of Human Diseases, The University of Texas Health Science Center at Houston, Houston, Texas, United States of America

☯ These authors contributed equally to this work.
* llwang@bcm.edu (LLW); dung-fang.lee@uth.tmc.edu (D-FL)

**Data Availability Statement:** The data supporting the findings of this study are available within the paper and its Supplementary Information. The RNA-seq data are available in the Gene Expression

## Abstract

Rothmund-Thomson syndrome (RTS) is an autosomal recessive genetic disorder characterized by poikiloderma, small stature, skeletal anomalies, sparse brows/lashes, cataracts, and predisposition to cancer. Type 2 RTS patients with biallelic *RECQL4* pathogenic variants have multiple skeletal anomalies and a significantly increased incidence of osteosarcoma. Here, we generated RTS patient-derived induced pluripotent stem cells (iPSCs) to dissect the pathological signaling leading to RTS patient-associated osteosarcoma. RTS iPSC-derived osteoblasts showed defective osteogenic differentiation and gain of *in vitro* tumorigenic ability. Transcriptome analysis of RTS osteoblasts validated decreased bone morphogenesis while revealing aberrantly upregulated mitochondrial respiratory complex I gene expression. RTS osteoblast metabolic assays demonstrated elevated mitochondrial

Omnibus (GEO) repository under accession number GSE174720. The WES data are available at the sequencing read archive (SRA) under accession number PRJNA768889.

**Funding:** B.E.J. and R.Z. were supported by the CPRIT UTHealth Innovation for Cancer Prevention Research Training Program Predoctoral Fellowship (RP160015). B.E.J. and N.K.B. were supported by the Gulf Coast Consortia, Training Interdisciplinary Pharmacology Scientists (TIPS) Program (NIH T32GM120011), and John J. and Charlene Kopchick Fellowships. D.Z. was supported by Congressionally Directed Medical Research Programs/DoD Horizon Award (W81XWH-20-1-0389). A.X. was supported by CPRIT UTHealth BIG-TCR training program. J.T. and Z.H. were supported by the Ke Lin Program of the First Affiliated Hospital of Sun Yat-sen University. D.-F.L. was supported by CPRIT RR160019, NIH/NCI R01CA246130, and Pablove Foundation childhood cancer research grant (690785). D.-F.L. is a CPRIT Scholar in Cancer Research. D.-F.L. and L.L.W. were supported by the Rolanette and Berdon Lawrence bone disease program of Texas. L.L.W. was supported by the BCM Curtis and Doris K. Hankamer Foundation Collaborative Research Grant, Amschwand Sarcoma Cancer Foundation, Kurt Groten Family Research Scholar's Program, Gillson Longenbaugh Foundation, Doris Duke Charitable Foundation Clinician Scientist Development Program, Eunice Kennedy Shriver National Institute of Child Health & Human Development (HD42136), Rothmund-Thomson Syndrome Foundation and CPRIT RR000188-42. Portions of this work were supported by National Institutes of Health (RR000188-42) to the Baylor College of Medicine General Clinical Research Center, and (HD083092) to the Clinical Translational Core and Tissue Culture Core of the Baylor College of Medicine Intellectual and Developmental Disabilities Research Center to L.L. W. B.A.K. was supported by NCI R01CA253445, R01CA234479 and Congressionally Directed Medical Research Programs W81XWH-18-1-0714. Z.Z. was partially supported by NIH R01LM012806. H.K.E was supported by NIH grants R01HL154720, R01DK122796, R01HL133900, R01DK109574, and Congressionally Directed Medical Research Programs /Department of Defense grant W81XWH2110032. N.R. was supported by Snowdrop Foundation. The funders had no role in study design, data collection and analysis, decision to publish, or preparation of the manuscript.

**Competing interests:** The authors have declared that no competing interests exist.

respiratory complex I function, increased oxidative phosphorylation (OXPHOS), and increased ATP production. Inhibition of mitochondrial respiratory complex I activity by IACS-010759 selectively suppressed cellular respiration and cell proliferation of RTS osteoblasts. Furthermore, systems analysis of IACS-010759-induced changes in RTS osteoblasts revealed that chemical inhibition of mitochondrial respiratory complex I impaired cell proliferation, induced senescence, and decreased MAPK signaling and cell cycle associated genes, but increased H19 and ribosomal protein genes. In summary, our study suggests that mitochondrial respiratory complex I is a potential therapeutic target for RTS-associated osteosarcoma and provides future insights for clinical treatment strategies.

## Author summary

Rothmund-Thomson syndrome (RTS) is an autosomal recessive genetic disease characterized by an array of clinical phenotypes affecting multiple tissues. Type 2 RTS is caused by pathogenic variants in the *RECQL4* gene encoding the RECQL4 DNA helicase. Type 2 RTS patients are prone to developing multiple primary osteosarcomas and have limited chemotherapy options due to organ toxicities or lifetime limits on active agents such as anthracyclines. There is currently no available RTS model that recapitulates the bone malignancy phenotype in this disease, severely limiting the ability to explore new treatment avenues which are greatly needed for these patients. To overcome this problem, we established a Type 2 RTS disease model using a human induced pluripotent stem cell platform. We then applied an unbiased approach to explore novel molecular mechanisms involved in *RECQL4* mutation-induced osteosarcoma to explore therapeutic interventions. Our findings indicate that mitochondrial respiratory complex I is an "Achilles' heel" of RTS osteosarcoma and that cancers harboring *RECQL4* mutations/deletions, in general, may be vulnerable to mitochondrial respiratory complex I inhibition.

## Introduction

Rothmund-Thomson syndrome (RTS) is a rare autosomal recessive disorder first described by Dr. Auguste Rothmund in 1868 in his patients who had the characteristic poikilodermatous skin rash and bilateral juvenile cataracts. In 1921 Dr. Sydney Thomson reported a similar condition but in association with radial ray defects [1–3]. The clinical phenotype of RTS was subsequently further delineated to include other features including growth retardation, hair, nail, and skeletal abnormalities, and a predisposition to cancer, specifically osteosarcoma [1–3]. Pathogenic variants in the *ANAPC1* and *RECQL4* genes have been identified in Type 1 (OMIM #618625) and Type 2 (OMIM #268400) RTS patients, respectively [4,5]. *RECQL4* pathogenic variants associated with RTS include nonsense, frameshift, splicing, and intronic deletions, all of which impair RECQL4 cellular function. In contrast to Type 1 RTS patients, who have no elevation in risk of osteosarcoma, Type 2 RTS patients with biallelic *RECQL4* pathogenic variants have a significantly elevated lifetime risk of osteosarcoma. While the clinicopathogical features of RTS-associated osteosarcomas are similar to osteosarcomas arising in the general population [6], they occur at an earlier age, and RTS patients are at risk of developing multiple primary tumors.

One hallmark feature of cancer cells is their ability to metabolically reprogram mitochondrial metabolism to support continued growth [7]. The mitochondria are central cellular

organelles responsible for metabolizing carbohydrates, lipids, and amino acids and for generating over 90% of cellular ATP and metabolic intermediates such as FADH2, NADH, and NADPH for the electron transport chain (ETC) and oxidative phosphorylation (OXPHOS). Cancer cells may facilitate growth by reprogramming their energy metabolism to favor several pathways, including acquisition of aerobic glycolysis (Warburg effect), increase in fatty acid oxidation, enhancement of glutamine and amino acid metabolism, and activation of OXPHOS [8,9]. Although the majority of cancer cells preferentially employ aerobic glycolysis to meet anabolic demands, some cancer cells rely uniquely on OXPHOS to support cell proliferation and survival [10]. In addition to promoting tumor proliferation, OXPHOS also drives tumor migration, metastasis, and immortalization [11,12]. These findings indicate the potential of therapeutically targeting OXPHOS-dependent tumors via their mitochondrial functions.

Although immortalized cancer cell lines, primary patient samples and cultured cells, and small organism models (e.g., fruit fly, zebrafish, and mouse) have made innumerable contributions to our understanding of the molecular mechanisms of cancer initiation, progression, metastasis, and recurrence, the complexity of the cancer genome and differences among species frequently limit clinical translation of many preclinical findings. Patient-derived induced pluripotent stem cells (iPSCs) provide unparalleled advantages as a model system, allowing investigators to study a cell continuously from the moment it differentiates from a multipotent progenitor into a differentiated cell type of interest [13,14]. Cancer disease modeling using patient-derived iPSCs has been employed successfully to study malignancies driven by a defined genetic background [15,16], including osteosarcoma in Li-Fraumeni syndrome (LFS) patients [17–20], leukemia in myelodysplastic syndrome (MDS) and Noonan syndrome patients [21,22], and colorectal cancers in familial adenomatous polyposis (FAP) patients [23]. These findings illustrate the feasibility of using patient-derived iPSCs as a platform to elucidate the pathological mechanisms involved in tumor initiation and development.

In this study, we established RTS iPSCs to analyze the transcriptomic alterations in RTS osteoblasts and to determine the pathological signature involved in RTS-associated osteosarcoma. We then applied these findings through multiple *in vitro* assays to identify a treatment strategy with promising activity against *RECQL4*-mutated osteoblasts. Our study has the potential to offer a paradigm shift in the treatment of osteosarcoma in RTS patients and *RECQL4*-mutated osteosarcoma. Suppression of mitochondrial respiratory complex I by OXPHOS inhibitor may offer effective salvage therapies, which currently do not exist for RTS patients as well as for recurrent sporadic osteosarcoma patients with somatic *RECQL4* mutations.

## Results

### Generation and characterization of iPSCs from RTS patients and unaffected parents

To gain insight into RTS-associated bone malignancies, we employed a patient-derived iPSC disease platform constructed from a cohort of four individuals from two RTS families, each with one affected RTS proband and one unaffected parent. Both RTS probands (labeled as RTS-A and RTS-B) had biallelic *RECQL4* pathogenic variants (Fig 1A), while parents (labeled as Family control A (FC-A) and Family control B (FC-B)) each had a *RECQL4* variant on one allele. Both RTS probands presented with classic Type 2 RTS features, including poikiloderma, radial ray defects, growth retardation, multiple skeletal and dental abnormalities (see Fig 1B for photos and Table 1 for clinical features), and both developed osteosarcomas. The *RECQL4* pathogenic variants in RTS patients were confirmed by Sanger sequencing (Fig 1C). Immuno-blotting indicated the loss of full-length RECQL4 protein in both RTS-A and RTS-B cells, suggesting impairment of RECQL4 function in these RTS patients (Fig 1D).

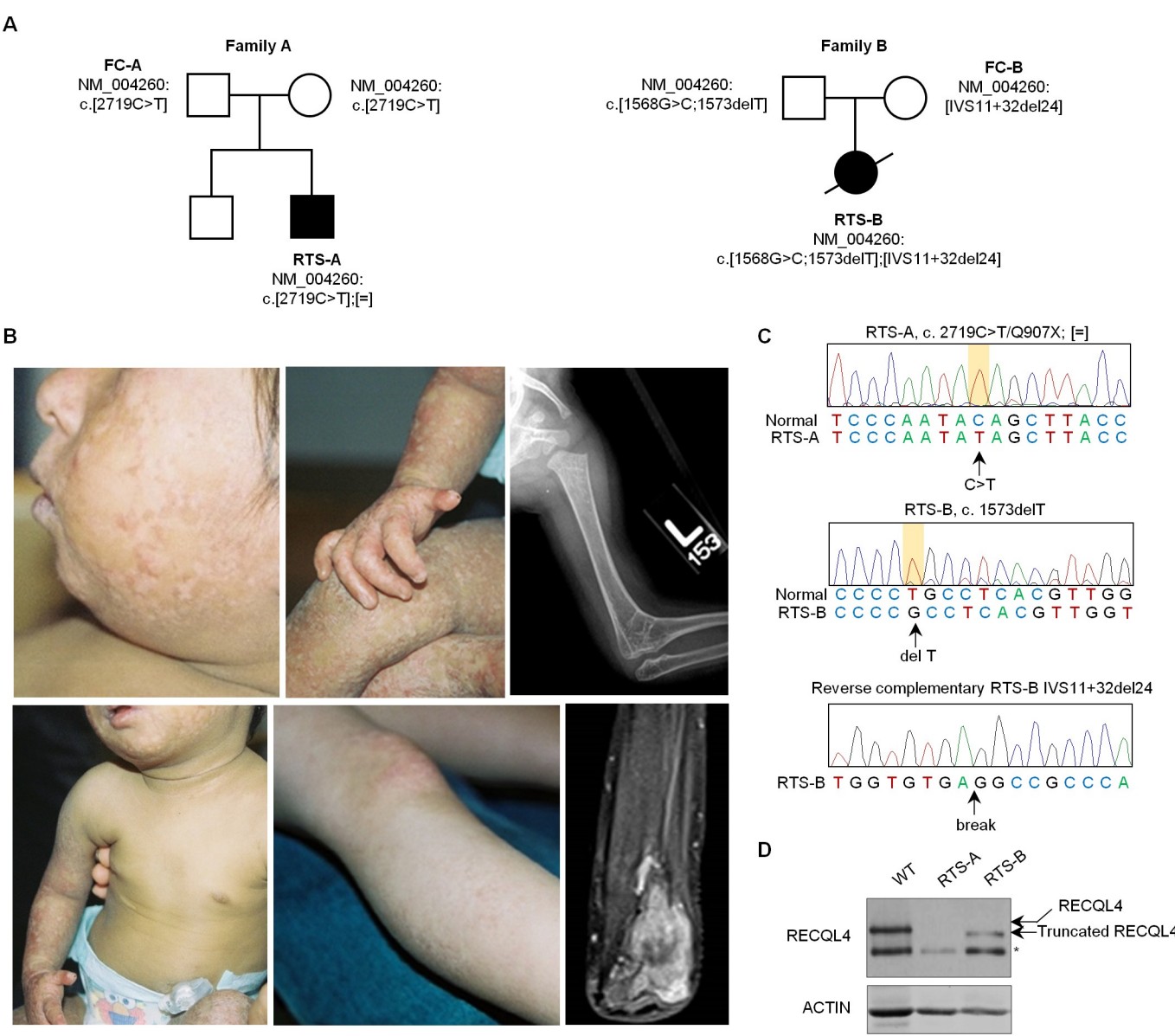

**Fig 1. Clinical characterization of Type 2 Rothmund-Thomson Syndrome patients in this study.** (A) Pedigrees of the two families with RTS Type 2 and *RECQL4* pathogenic variants reported in this study. (B) Photos show the classic poikiloderma on the face, ears, and arms of individual RTS-A at age 2 years (left panels, top and bottom). Note the sparing of the trunk and abdomen as well as sparse scalp hair and absence of eyebrows and presence of gastrostomy feeding tube. Middle panels illustrate variability in the severity of poikiloderma on the lower extremities in RTS-A (upper) and RTS-B (lower). Note hypoplastic thumb and dystrophic nails in RTS-A. Both individuals developed osteosarcoma. The right top panel shows an x-ray of the skeletal defect (radiohumeral synostosis, age 2 years) in RTS-A, and the bottom panel shows an MRI image of osteosarcoma that developed in the right proximal radius of RTS-A at age 10 years. This T1 fat-saturated, coronal, post-contrast view demonstrates diffuse solid enhancement of the tumor with some central areas of non-enhancement. (C) Sanger sequencing verifies biallelic *RECQL4* (c. 2719C>T/Q907X) pathogenic variant in RTS-A fibroblasts (upper panel) and *RECQL4* (1568G>C;1573delT and VS11 +32del24) in RTS-B fibroblasts (middle and lower panels). (D) Immunoblotting indicates the loss of RECQL4 protein in RTS-A fibroblasts and the truncated RECQL4 protein in RTS-B fibroblasts. *, non-specific band.

Fibroblasts from RTS individuals and their parents were transduced with four Yamanaka factors (NANOG, OCT4, SOX2, and MYC) [24,25] using a Sendai virus (SeV)-based cell reprogramming method. iPSC clones were successfully isolated and expanded from fibroblasts, leading to generation of iPSC clones from two affected RTS probands (RTS-A and RTS-B) and two parents (FC-A and FC-B). The iPSCs exhibited hESC clonal morphology and

**Table 1. Clinical characteristics of RTS Type 2 individuals with RECQL4 mutations.**

|  | RTS-A | RTS-B |
|---|---|---|
| **Gender** | M | F |
| **Skin** | Poikiloderma, hyperkeratosis | Poikiloderma |
| **Hair** | Sparse scalp hair, absent eyebrows, eyelashes | Sparse scalp hair, absent eyebrows, eyelashes |
| **Eyes** | Astigmatism, epiphoria | Astigmatism, epicanthal folds |
| **ENT** | Recurrent OM, mild hearing loss, mild nasal and choanal stenosis | Normal |
| **Teeth** | Dental crowding | Dental caries, erosion, enamel defects |
| **Nails** | Dystrophic nails | Normal |
| **Skeletal** | Bilateral thumb and digit hypoplasia, left radiohumeral synostosis, metaphyseal trabeculation defects; osteopenia | Bilateral thumb and digit hypoplasia, left radius hypoplasia, limited elbow extension |
| **Growth and Development** | Small stature, speech and motor delay | Small stature, delayed milestones |
| **GI** | History of vomiting diarrhea, malabsorption, feeding tube | History of severe vomiting as infant |
| **Cancer** | Osteosarcoma, diagnosed at 10 years | Osteosarcoma, diagnosed at 11 years |

Abbreviations: M-male; F-female; ENT-Ear, Nose Throat; OM-otitis media; GI-Gastrointestinal

ubiquitously expressed the pluripotency transcription factors NANOG and OCT4 as well as human embryonic stem cell (hESC) surface antigens TRA-1-81 and SSEA4 (Fig 2A). Quantitative reverse transcription PCR (qRT-PCR) confirmed comparable expression of *NANOG*, *OCT4*, and *SOX2* mRNA transcripts in iPSC lines compared to hESC H1 cells (Fig 2B). Teratoma formation assay was performed to test the ability of these cell lines to differentiate into all three germ layers *in vivo*. H&E examination of teratomas revealed successful differentiation of all generated lines into ectodermal, endodermal, and mesodermal lineages (S1A Fig), confirming their pluripotency. We further verified the loss of exogenous SeV-OCT4, SOX2, KLF4, and MYC transgenes by genomic PCR detection and confirmed absence of detectable SeV transgenes in these iPSCs (S1B Fig), indicating that these RTS and FC iPSCs are generated with zero genetic footprint. To further investigate if additional mutation burden was created during reprogramming, we applied whole exome sequencing (WES) to compare RTS fibroblasts and iPSCs. In comparison with RTS-B fibroblasts, 6 and 146 additional nonsynonymous somatic variants were detected in RTS-B2 and RTS-B3, respectively (S1C Fig and S1 Table). The number of somatic variants resembled the level of mutations found in reprogrammed iPSCs as previously reported [26]. No mutations related to known tumor suppressor genes and oncogenes (e.g., *p53*, *RB1, and* RAS*)* were detected in RTS iPSCs. Taken together, these iPSC characterizations suggested the reliability of the RTS iPSC platform to model and study RTS pathology.

## RTS iPSC-derived osteoblasts demonstrate osteogenic differentiation defects and premalignant phenotypes

We differentiated the newly generated iPSC lines to mesenchymal stem cells (MSCs) using an optimized MSC differentiation protocol [27]. During the 8-week MSC differentiation period, RTS and FC iPSCs exhibited morphological changes under brightfield microscopy from compact iPSCs to elongated and swirling cellular patterns characteristic of MSCs (Fig 2C). Immunostaining of these differentiated cells demonstrated presence of typical MSC surface markers CD44, CD73, and CD105 (Fig 2C), confirming successful differentiation. These MSCs were further differentiated into osteoblasts, a potential cell-of-origin for osteosarcoma, by a defined osteoblast differentiation method [17]. We further characterized cell proliferation of RTS and FC samples at different stages (iPSCs, MSCs, and osteoblasts) and found that RTS iPSCs and

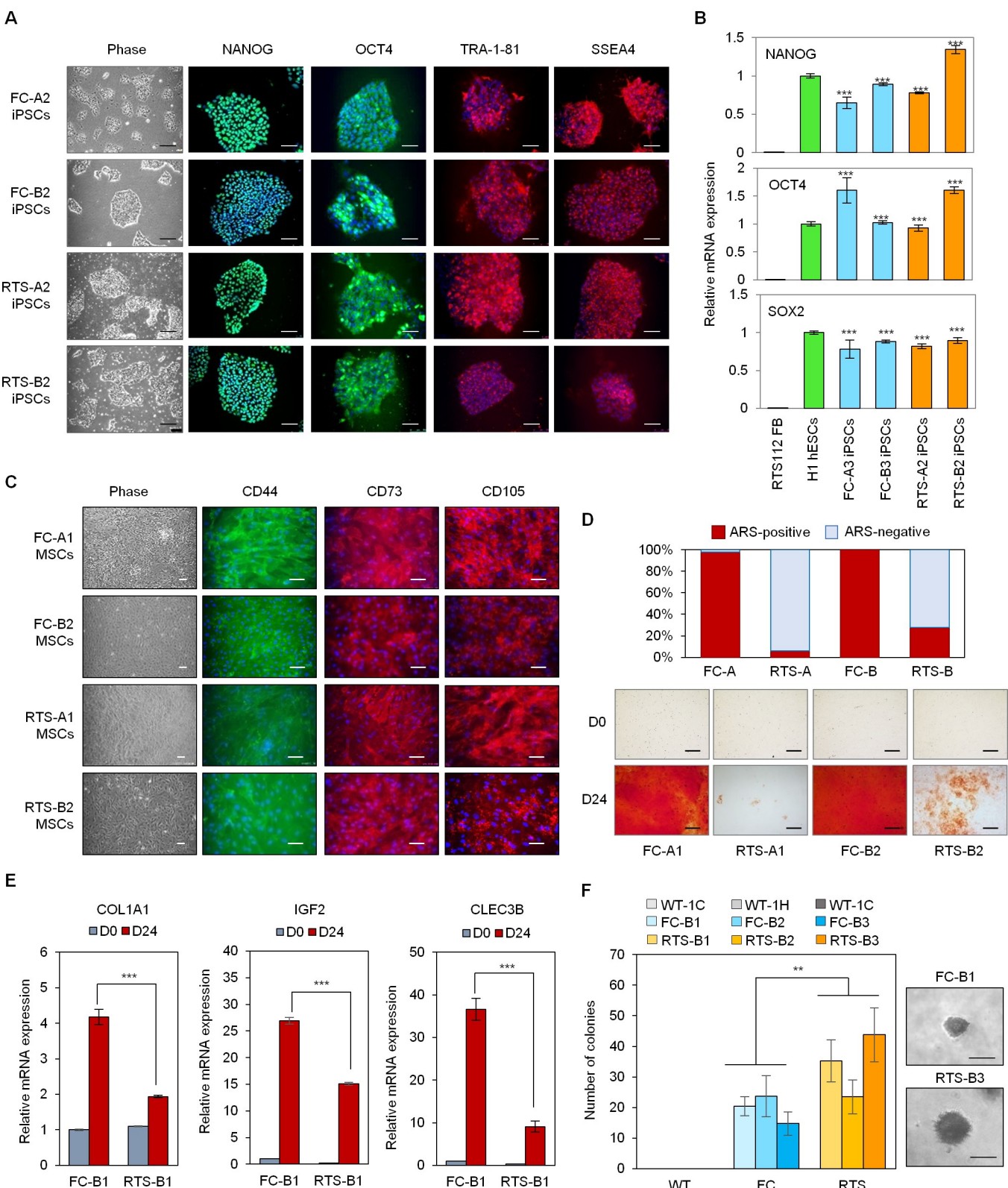

**Fig 2. Generation of RTS iPSCs and iPSC-derived osteoblasts.** (A) RTS and FC iPSCs exhibit hESC morphology and express pluripotency transcription factors NANOG and OCT4 as well as hESC surface antigens TRA-1-81 and SSEA44 ubiquitously. Scale bar, 100 μm. (B) qRT-PCR assay for pluripotency genes *NANOG*, *OCT4*, and *SOX2* in RTS and FC iPSCs. Error bars indicate SEM of triplicates. n = 3 biological replicates. (C) Immunostaining demonstrates that

iPSC-derived MSCs exhibit swirling morphology (phase contrast image) and express MSC surface markers CD44, CD73, and CD105. Scale bar, 100 μm. (D) ARS staining reveals attenuated mineral deposition ability in RTS osteoblasts. Scale bar, 100 μm. (E) qRT-PCR assay for osteoblast lineage genes *COL1A1*, *IGF2*, and *CLEC3B* in RTS and FC osteoblasts. Error bars indicate SEM of triplicates. n = 3 biological replicates. (F) AIG assay for *in vitro* tumorigenicity demonstrates that RTS osteoblasts present an increased clonal growth ability in soft agar in comparison with FC and WT osteoblasts. Positive colonies after 1.5-month growth of differentiated osteoblasts in osteoblast differentiation medium are those larger than 50 μm (scale bar, 50 μm). n = 3 biological replicates.

MSCs have decreased cell proliferation ability compared to FC counterparts, indicating the essential role of RECQL4 in maintaining cell growth (S2A Fig, upper and middle panels). Importantly, there were no significant growth differences between RTS and FC osteoblasts (S2A Fig, lower panel). Consistent with the role of RECQL4 in regulating the cell cycle [28], cell cycle profile analyses revealed that RTS iPSCs have an increased G2/M cell population, whereas RTS MSCs and osteoblasts do not show differences in their G2/M cell population (S2B Fig). These results imply potential pro-oncogenic transformation (e.g., enhanced cell growth ability) in RTS osteoblasts. Wound healing assays demonstrated no difference in cell migration abilities between RTS and FC osteoblasts (S2C Fig), which may reflect the clinical observation that RTS patient osteosarcomas do not have a prevalence of metastases [6]. Furthermore, Alizarin Red S (ARS) staining for mineral deposition by functional osteoblasts was decreased in RTS compared to FC samples (Fig 2D), suggesting impaired osteogenic differentiation in RTS osteoblasts. Consistently, in comparison with FC osteoblasts, RTS osteoblasts showed lower expression of the osteoblastic genes COL1A1, IGF2, and CLEC3B during osteogenesis (Fig 2E). Anchorage-independent growth (AIG) assay showed that RTS osteoblasts demonstrate an increased clonal growth ability in soft agar in comparison with FC and wild-type (WT) osteoblasts (lacking biallelic *RECQL4* pathogenic variants) (Fig 2F), supportive of an *in vitro* tumorigenic ability of RTS osteoblasts. Remarkably, FC osteoblasts could maintain a certain clonal growth ability in soft agar, implying a potential tumor suppressor role of RECQL4 at the tumor imitation stage. Taken together, these findings demonstrate that aspects of the Type 2 RTS phenotype, in particular abnormal skeletal development and premalignant ability, can be recapitulated within the RTS iPSC disease platform.

## Increased mitochondrial respiratory gene transcripts in RTS osteoblasts

To interrogate the molecular dysregulation leading to osteosarcomagenesis in RTS patients, RTS and FC iPSC-derived cells were collected for transcriptome analysis by RNA-seq at three different osteogenic differentiation stages: MSC stage (day 0; D0); pre-osteoblast stage (day 15; D15); and mature osteoblast stage (day 24; D24). We applied gene set enrichment analysis (GSEA) to identify gene ontology biological processes (GO_BPs) enriched in RTS or FC cells at these differentiation stages. GO_BPs for many mitochondrial energy production functions (e.g., ATP synthesis coupled electron transport, mitochondrial respiratory chain complex assembly, mitochondrial electron transport NADH to ubiquinone, NADH dehydrogenase complex assembly, OXPHOS, etc.) were enriched in RTS osteoblasts compared to FC osteoblasts (Fig 3A and 3B). Interestingly, enrichment of these pathways was only observed in osteoblast but not MSC lineages, indicating that the alteration of mitochondrial energy production occurs during, but not prior to, osteogenic differentiation. In contrast, GO_BP analyses revealed significantly depleted genes in skeletal development pathways (e.g., bone morphogenesis, bone development, cell matrix adhesion, extracellular matrix assembly, etc.) in the RTS pre-osteoblasts and osteoblasts compared to FC counterparts (Fig 3A and 3B), supportive of the small stature and skeletal abnormalities seen in RTS patients (Fig 1B and Table 1) and the impaired osteogenic differentiation observed in RTS osteoblasts (Fig 2D).

Consistently, KEGG pathway analyses highlighted increased mitochondrial ATP production pathways, including OXPHOS and tricarboxylic acid (TCA) cycle pathway genes, but

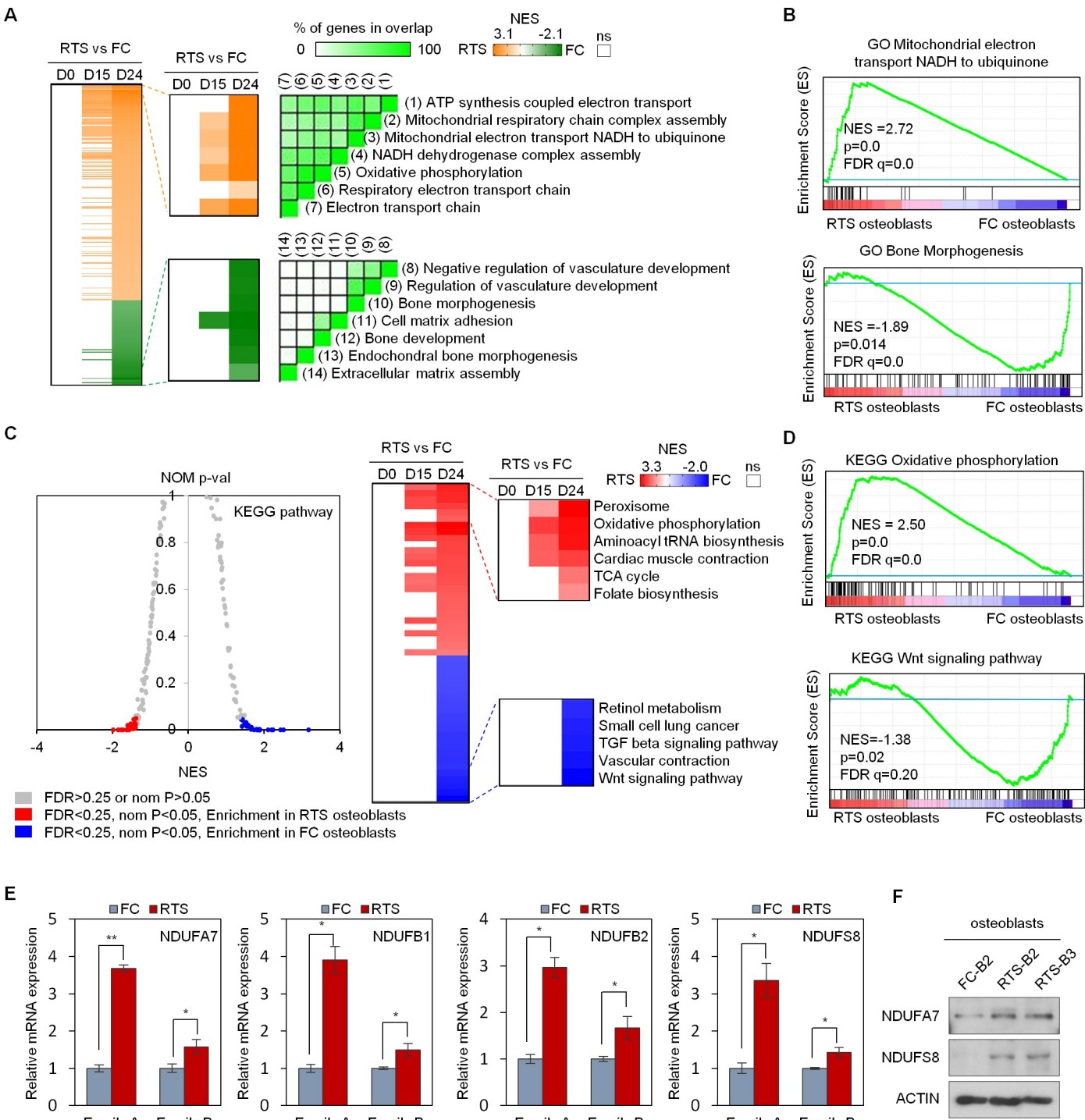

**Fig 3. Transcriptional analysis reveals increased OXPHOS in RTS osteoblasts.** (A) GSEA analyses identify gene ontology biological processes (GO_BP) from a collection of 7525 genes sets (left) enriched in RTS or FC osteoblasts. GO_BP gene sets enriched (orange, corresponding to a positive normalized enrichment score (NES)) or depleted (green, corresponding to a negative NES) in the transcriptome of RTS compared to FC osteoblasts are shown. Enriched gene sets are selected based on statistical significance (normalized p-value < 0.05 and FDR q-value < 0.25). Middle, heatmap of significantly altered GO_BPs in RTS MSCs (D0), pre-osteoblasts (D15), and osteoblasts (D24) compared to FC counterparts. Right, GSEA leading edge analysis demonstrates the overlap between gene sets enriched in RTS and FC osteoblasts. (B) GSEA GO_BP results for mitochondrial electron transfer NADH to ubiquinone and bone morphogenesis. (C) KEGG pathway analyses identify enriched pathways in RTS and FC osteoblasts. Left, 183 total pathways are included in KEGG pathway analysis. Pathways enriched in the transcriptome of FC osteoblasts have a negative normalized enrichment score (NES) (blue), while pathways with a positive NES (shown in red) are enriched in the transcriptome of RTS osteoblasts. Enriched pathways are selected based on statistical significance (normalized p-value < 0.05 and FDR q-value < 0.25). Right, heatmap showing

significantly altered pathways in RTS MSCs (D0), pre-osteoblasts (D15), and osteoblasts (D24) compared with FC counterparts. (D) Representative KEGG pathways differentially regulated in RTS osteoblasts, including OXPHOS and Wnt signaling pathway. (E) Individual genes (*NDUFA7*, *NDUFB1*, *NDUFB2*, and *NDUFS8*) that compose complex I of the electron transport chain are significantly upregulated in RTS osteoblasts. n = 3 biological replicates. (F) Immunoblotting indicates that NDUFA7 and NDUFS8 are upregulated in RTS osteoblasts compared to FC osteoblasts.

decreased osteogenic Wnt signaling genes in RTS compared to FC osteoblasts (Fig 3C and 3D). These findings were further supported by Reactome biologic pathways and process analyses showing an increase in transcription of genes related to respiratory electron transport and respiratory electron transport ATP synthesis but decreased transcription of genes related to extracellular matrix organization in RTS compared to FC osteoblasts (S3A Fig). We validated the upregulation of select OXPHOS genes in RTS osteoblasts using qRT-PCR and confirmed upregulation of *NDUFA7*, *NDUFB1*, *NDUFB2*, and *NDUFS8*, which are involved in both the accessory and catalytic subunits of mitochondrial respiratory complex I [29], in RTS compared to FC osteoblasts (Fig 3E). The upregulation of NDUFA7 and NDUFS8 was further confirmed by immunoblotting (Fig 3F). These findings are consistent with transcriptome analyses of osteosarcoma cell lines and specimens demonstrating increased mitochondrial ATP production pathways (e.g., OXPHOS, respiratory electron transport chain, TCA cycle, and mitochondrial respiratory complex assembly) in sporadic osteosarcoma cell lines compared with normal bone tissues and osteoblasts (S3B Fig), suggesting that elevated expression of mitochondrial ATP production genes contributes to osteosarcomagenesis.

## Increased mitochondrial respiratory complex I activity in RTS osteoblasts facilitates elevated OXPHOS

To explore the biochemical activity of the electron transport chain in RTS and FC osteoblasts, we systematically analyzed the enzymatic activities of the major mitochondrial respiratory protein complexes including complex I, complex I and III, complex II, complex II and III, and citrate synthase enzyme activity. We discovered that RTS osteoblasts had significantly increased mitochondrial respiratory complex I activity (Fig 4A), but no difference in complex II, III, IV, or citrate synthase enzyme activities (S4A–S4E Fig). These mitochondrial enzymatic activity assays were consistent with transcriptome results showing increased mRNA expression of mitochondrial respiratory complex I genes *NDUFA7*, *NDUFB1*, *NDUFB2*, and *NDUFS8* (Fig 3E). These findings lead us to conclude that RTS osteoblasts are characterized by increased mitochondrial respiratory complex I gene transcription and activity.

Mitochondrial respiratory complex I is of particular importance as it is the first step in the electron transport chain, establishing the electrochemical gradient necessary for ATP production [30]. Next, we explored the cellular metabolism of RTS osteoblasts using Seahorse metabolic flux assays. Our results demonstrated an increased oxygen consumption rate (OCR) in RTS osteoblasts compared to FC osteoblasts (Fig 4B). The basal respiration rate of RTS osteoblasts, as measured by ATP production, was also significantly increased compared to FC osteoblasts (Fig 4C). RTS osteoblasts also had significantly increased proton leak under normal culture conditions (Fig 4D), indicating an increased ability to create the proton gradient across the mitochondrial inner matrix for ATP synthase activity. Indeed, RTS osteoblasts showed increased ATP-linked (i.e., mitochondrial) respiration and maximal respiration (Fig 4E and 4F). RTS osteoblasts also retained significantly more spare respiratory capacity (Fig 4G). In contrast, RTS and parental osteoblasts had similar non-mitochondrial respiration rates (Fig 4H), suggesting that mitochondria-associated OXPHOS is the cause of increased oxygen consumption.

We next compared extracellular acidification rates (ECAR) to assay for glycolysis in RTS osteoblasts. In agreement with their elevated OXPHOS, RTS osteoblasts shift away from

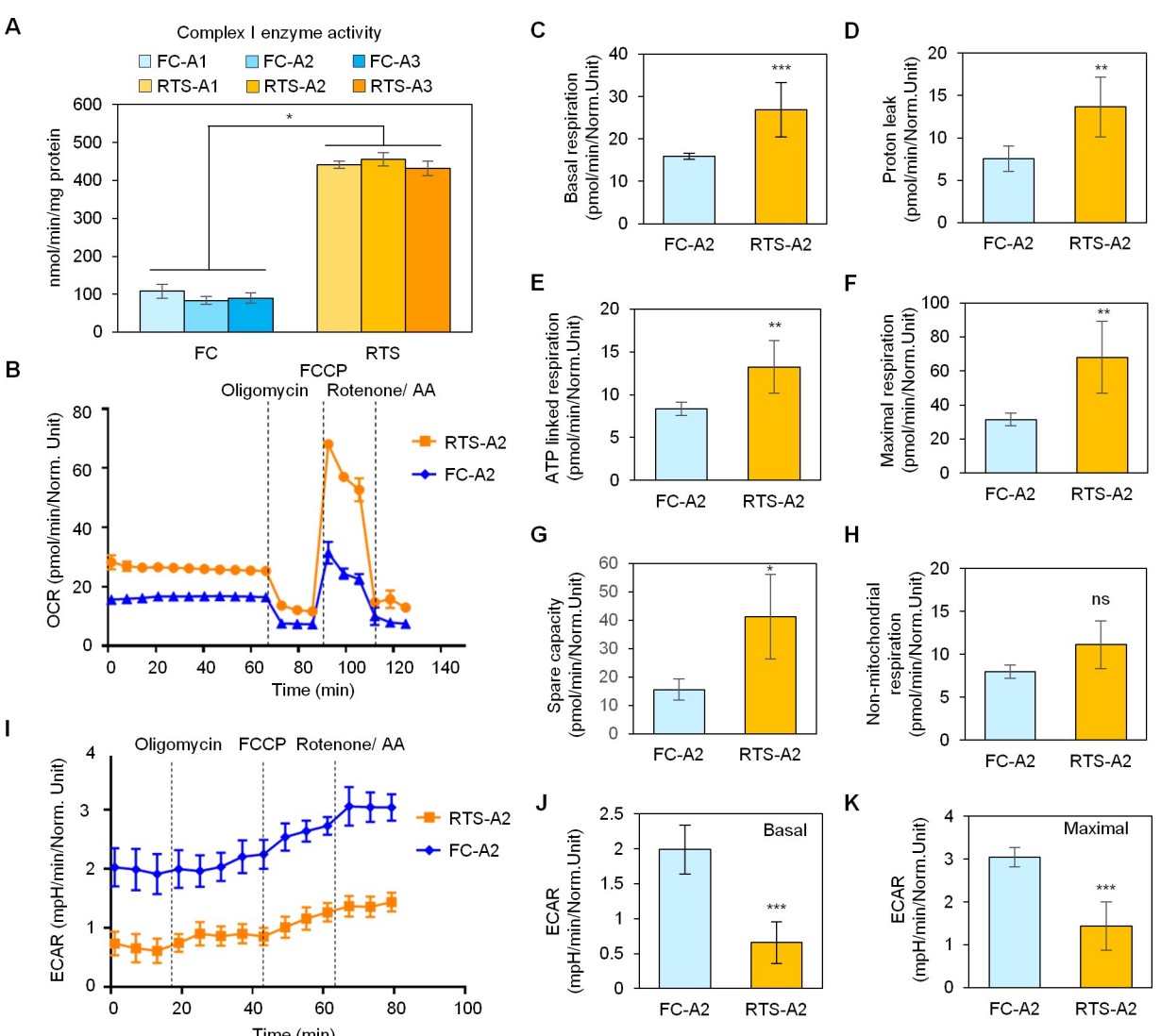

**Fig 4. Metabolic differences between RTS and FC osteoblasts demonstrate the role of OXPHOS in increasing ATP production in RTS osteoblasts.** (A) Enzyme activity of complex I is increased in RTS osteoblasts. n = 3 biological replicates. (B) Seahorse assays indicate increased oxygen consumption rate, a measure of OXPHOS, in RTS osteoblasts. n = 3 biological replicates. (C-H) Individual calculations of basal respiration (C), proton leak (D), ATP-linked respiration (E), maximal respiration (F), and spare capacity (G) in RTS osteoblasts as measured by seahorse assay, all of which are significantly upregulated. Non-mitochondrial respiration (H), which is not attributed to OXPHOS, is comparable between RTS and parental osteoblasts. n = 3 biological replicates. (I) Seahorse assays show decreased extracellular acidification rate, a measure of glycolysis, in RTS osteoblasts. n = 3 biological replicates. (J-K) Significantly decreased basal extracellular acidification rate (J) and maximal acidification rate (K) in RTS osteoblasts indicate RTS osteoblast reliance on OXPHOS for ATP production. n = 3 biological replicates.

glycolysis toward OXPHOS as a means of ATP production (Fig 4I) and demonstrated a significant decrease in both basal and maximal ECAR (Fig 4J and 4K), indicating that RTS osteoblasts have little reliance on glycolytic metabolism. Also, a decrease of total ATP levels was found in RTS osteoblasts (S4F Fig). These data demonstrate that, compared to FC osteoblasts, RTS osteoblasts upregulate OXPHOS and mitochondrial respiratory complex I -mediated ATP production at the expense of decreased glycolysis. Taken together, our systems-based analyses demonstrating an increase in mitochondrial respiratory transcripts and activities lead us to hypothesize that elevated complex I enzymatic activity in RTS osteoblasts may play a critical role in RTS-driven osteosarcomagenesis.

## IACS-010759 selectively inhibits OXPHOS and cell viability of RTS osteoblasts

OXPHOS inhibitor IACS-010759 binds to and inhibits mitochondrial respiratory complex I of the electron transport chain, thereby selectively inhibiting growth and survival of cells dependent on complex I function [31,32]. A preclinical study of IACS-010759 demonstrated robust suppression of proliferation and induction of cell death in tumor cells reliant on OXPHOS. This agent is being evaluated in phase I clinical trials against acute myeloid leukemia (AML) and solid tumors [31]. Based on our results highlighting that RTS osteoblasts meet their increased energy demands via hyperactivated mitochondrial respiratory complex I function, we investigated whether RTS premalignant osteoblasts are selectively vulnerable to inhibition of mitochondrial respiratory complex I function. We first examined the effect of IACS-010759 on ATP production via OXPHOS in RTS osteoblasts. IACS-010759 treatment resulted in decreased OXPHOS (Fig 5A), with significantly reduced ATP-linked respiration, maximal respiration, and spare respiratory capacity in these cells (Fig 5B). We then assayed osteoblast growth following IACS-010759 inhibition. Upon IACS-010759 treatment, RTS osteoblasts, but not FC osteoblasts, altered their cell morphology from a spindle shape into an irregular shape (Figs 5C and S5A), indicating cellular effects on RTS osteoblasts. Consistently, cell proliferation assays demonstrated selectively inhibited RTS osteoblast growth following IACS-010759 treatment but only limited effects on FC osteoblast proliferation (Fig 5D), suggesting that inhibition of mitochondrial respiratory complex I activity selectively suppresses RTS cell survival. In contrast, Rotenone, another known mitochondrial respiratory complex I inhibitor, not only presented less efficacy in inhibiting RTS osteoblasts compared to FC counterparts but also showed a certain inhibitory effect on FC osteoblast growth (S5B Fig). These findings suggest that IACS-010759 is a better choice for clinical application.

We next investigated the cellular effects of IACS-010759-mediated RTS osteoblast growth inhibition. As shown in S5C Fig, RTS osteoblasts exhibited an increased reactive oxygen species (ROS) production compared to FC osteoblasts, but IACS-010759 did not change ROS levels in both RTS and FC osteoblasts. Caspase 3/7 activity assays indicated that IACS-010759 leads to decreased cell apoptosis in both RTS and FC osteoblasts. (S5D Fig). Interestingly, IACS-010759 selectively induced cellular senescence of RTS osteoblasts (Fig 5E) and altered the cell cycle profile of RTS osteoblasts by increasing G0/G1 and decreasing S and G2/M cell populations (S5E Fig). These findings suggest that induction of senescence, but not ROS production and apoptosis, is one of the pharmacological mechanisms involved in IACS-010759-induced RTS osteoblast growth inhibition. Taken together, these results demonstrate that inhibition of mitochondrial respiratory complex I abrogates OXPHOS and induces cell senescence in RTS osteoblasts and underscore the possibility of targeting RTS osteosarcomas using IACS-010759.

To shed light on the molecular effects of IACS-010759 in halting RTS osteoblast viability, we next examined the transcriptional changes in RTS osteoblasts upon IACS-010759 treatment. A scatter plot comparing gene expression between IACS-010759 and DMSO treated RTS osteoblasts indicated that IACS-010759 treatment upregulates long non-coding RNA H19 and numerous ribosome proteins (e.g., RSP27, RPL11, RPL32, RPL35A, RPL11, etc.) and downregulates cell cycle processes (e.g., EPGN, EREG, GPSM2, etc.) and the MAP kinase pathway (STK39, HGF, GHR, etc.) in these cells (Fig 5F). Enrichr-based GO_BP analysis of differentially expressed genes (>1.5 fold changes) between DMSO and IACS-010759-treated RTS osteoblasts demonstrated an increase in protein targeting, T cell mediated cytotoxicity, and protein translation pathway transcripts and a decrease in cell cycle process and MAP kinase (MAPK) pathway transcripts in IACS-010759-treated RTS osteoblasts (Figs 5G

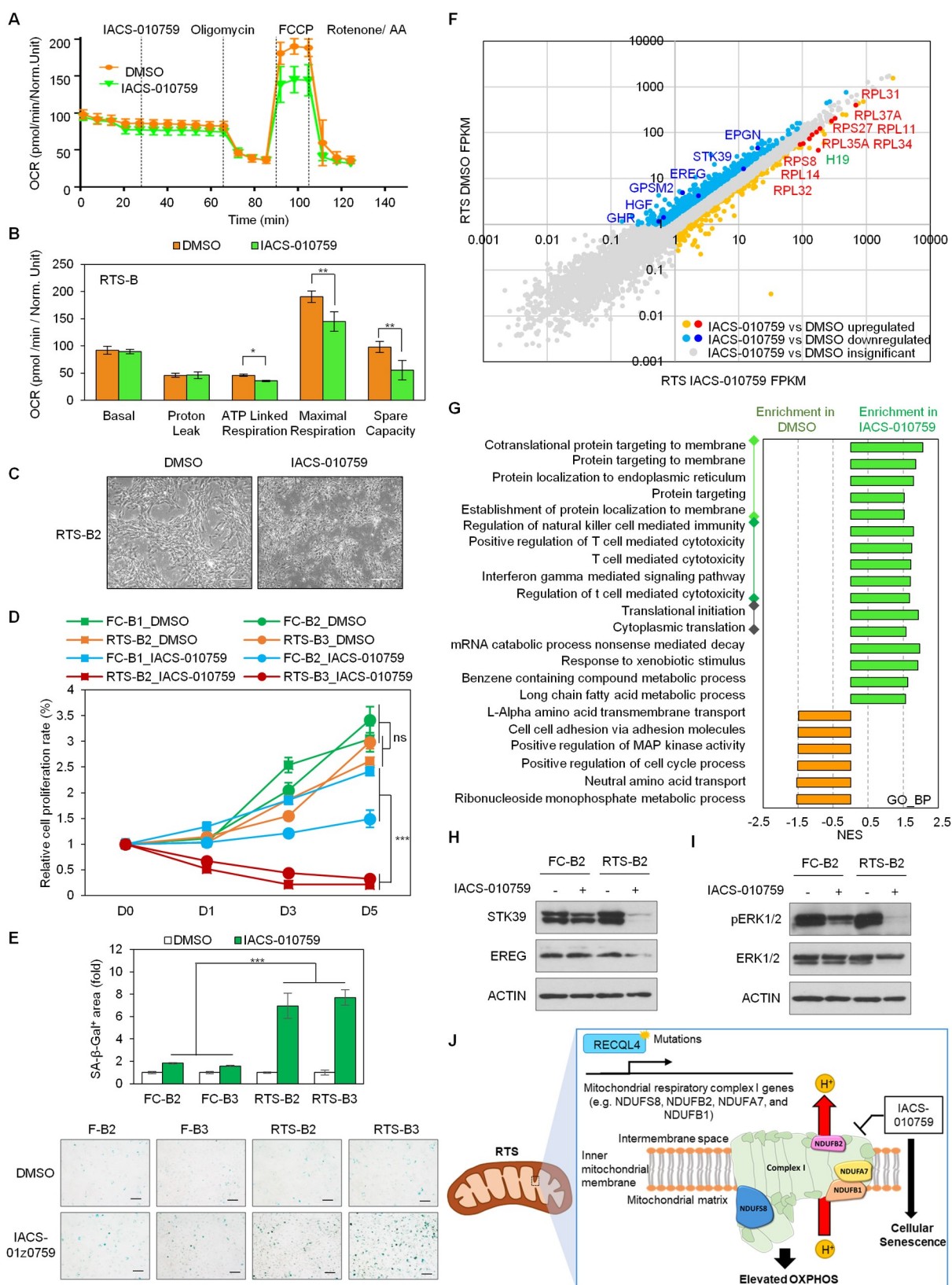

**Fig 5. Inhibition of mitochondrial respiratory complex I specifically impairs ATP respiration, reliance on OXPHOS, and survival in RTS osteoblasts.** (A-B) RTS osteoblasts treated by 100 nM of IACS-010759 show reduced ATP respiration, maximal respiration, and spare

capacity. Differences in OCR with seahorse assay are most apparent after the addition of FCCP, which measures the maximum potential of ATP production through the electron transport chain. n = 3 biological replicates. (C) IACS-010759 alters RTS osteoblast cell morphology from a spindle shape to an irregular shape. Scale bar, 100 μm. (D) IACS-010759 selectively inhibits cell proliferation of RTS osteoblasts. Cell proliferation assays after treatment with 100 nM of IACS-010759 show growth inhibition of RTS osteoblasts but little effect on FC osteoblasts. n = 4 biological replicates. (E) IACS-010759 induces the cellular senescence of RTS osteoblasts. The SA-β-Gal+ areas are measured using ImageJ. n = 3 biological replicates. (F) Scatter plot indicates a marked difference of transcripts between DMSO and 100 nM of IACS-010759-treated RTS osteoblasts. H19 and numerous ribosomal protein genes are upregulated but MAPK pathway and cell cycle-associated genes are downregulated upon IACS-010759 treatment. (G) GO_BP analysis indicates dysregulated pathways in RTS osteoblasts upon IACS-010759 treatment. IACS-010759 activates protein targeting, T cell-mediated toxicity, and protein translation related biological processes but inhibited MAPK and cell cycle processes in RTS osteoblasts. (H) Immunoblotting indicates a decrease MAPK pathway related STK9 and EREG in RTS osteoblasts upon IACS-010759 treatment. (I) IACS-010759 impairs ERK1/2 activity in RTS osteoblasts. (J) Model for the elevated mitochondrial respiratory complex I function in RTS osteoblasts. In RTS patients, impaired RECQL4 function leads to an increase of mitochondrial respiratory complex I gene expression and function, and thereby induces premalignant phenotypes. Pharmacological inhibition of mitochondrial respiratory complex I function by IACS-010759 can be applied to prevent and/or treat RTS patient associated osteosarcomas.

and S5F). Downregulation of cell cycle regulator EREG and MAPK signaling molecule STK39 (Fig 5H) and ERK1/2 activity (Fig 5I) in IACS-010759-treated RTS osteoblasts were validated by immunoblotting. KEGG pathway analysis of significantly upregulated genes in IACS-010759-treated RTS osteoblasts further validated the upregulation of the ribosome pathway (S5G Fig). In summary, our systematic transcriptome analyses revealed multiple potential pharmacologic mechanisms for IACS-010759-mediated cell growth inhibition in RTS osteoblasts.

## Discussion

In this study, we established an RTS iPSC disease model to investigate altered pathological mechanisms involved in RTS patient-associated osteosarcoma. By differentiating RTS iPSCs to osteoblasts and characterizing them along this pathway of differentiation, we were able to establish specific osteoblastic defects induced by the *RECQL4* genotypes that correlate with the RTS Type 2 phenotype. The advantage of the iPSC platform lies in its ability to establish highly flexible pluripotent human cell lines from an established unique genetic alteration and proven background. In this case, our approach allowed us to dissect the impact of *RECQL4* pathogenic variants on RTS-associated pathological consequences such as defective osteogenesis and osteosarcomagenesis. The unique self-renewal/pluripotency and differentiation abilities of iPSCs distinguish them from all existing model systems. In addition to MSCs and osteoblasts, RTS iPSCs are capable of being differentiated into other germ layers. With appropriate differentiation protocols, these models can be used to study other clinical phenotypes of RTS, such as cataracts, poikiloderma, and abnormal dentition, that affect other tissues and organ systems.

We examined the dysregulated pathways in RTS iPSC-derived osteoblasts using multiple gene set categories including GO_BP, KEGG pathway and Reactome. All tools for systems analyses pointed to the same alteration of increased energy metabolism and upregulated OXPHOS in RTS osteoblasts, indicating that *RECQL4* pathogenic variants result in elevated OXPHOS in RTS osteoblasts. Accumulating studies of energy metabolism in leukemias, lymphomas, pancreatic ductal adenocarcinoma, melanoma, and endometrial carcinoma reveal the essential role of OXPHOS in contributing to tumor formation and progression [10]. Our transcriptomic analyses also identified increased expression of genes involved in OXPHOS, particularly mitochondrial respiratory complex I, in RTS osteoblasts. Consistently, bioenergetics in RTS osteoblasts revealed the preference of RTS osteoblasts for utilizing OXPHOS but not aerobic glycolysis to meet anabolic demands. Since total ATP production is decreased in RTS cells (S4F Fig), an increase of OXPHOS by upregulating mitochondrial respiratory complex I gene expression to enhance ATP production would provide an essential advantage to overcome

ATP deficiency due to *RECQL4* pathogenic variants. Increased production of NAD+ through mitochondrial respiratory complex I has been shown to immortalize tumor cells at the tumor initiation stages [11]. We speculate that elevated mitochondrial respiratory complex I genes and OXPHOS provide dual RTS survival benefits, including energy for a higher level of growth as well as increased NAD+ to drive RTS osteoblast immortalization toward malignancy.

As a DNA helicase, RECQL4 not only plays an important role in modulating DNA replication and repair but also preserves mitochondrial DNA integrity. RECQL4 was suggested to interact with mitochondrial polymerase (PolγA/B2) and play a role in the proofreading activities of PolγA/B2 [33]. In agreement with elevated ROS in RTS osteoblasts (S5C Fig), RECQL4 homozygous deficient fibroblasts from RTS patients exhibited increased oxidative stress and a markedly decreased repair capacity for DNA damage [34]. Mutant RECQL4 particularly translocates from nucleus to mitochondria and then interacts with mitochondrial replication helicase PEO1, which leads to elevated mtDNA synthesis and mitochondrial dysfunction [35]. Our study added a new finding to link mitochondrial dysfunction to RTS pathology and uncovered the upregulated mitochondrial respiratory complex I gene expression and activities in RTS osteoblasts. Together, these findings shed light on additional RECQL4 functions in mitochondria and the involvement of mitochondria dysregulation in RTS etiology.

Our thorough transcriptional and biochemical studies conclude that elevated mitochondrial respiratory complex I function in RTS osteoblasts is a potential therapeutic target to treat RTS osteosarcoma. Indeed, the preclinical compound IACS-010759, a mitochondrial respiratory complex I inhibitor specific for the ND1 subunit that blocks the intermembrane transport of H+ [31], selectively impairs cell viability of RTS osteoblasts. Our RTS osteoblast transcriptome analyses demonstrated that IACS-010759 treatment not only impairs MAPK signaling and cell cycle gene expression, but also upregulates H19 and ribosomal gene expression. The effect of the RTS genotype on the MAPK pathway has been previously shown in RTS fibroblasts, which demonstrated an increased cellular lifespan and proliferation rate due to MAPK activation [36]. H19 has also been shown to function as a tumor suppressor to promote osteogenesis and inhibit osteosarcomagenesis in LFS osteoblasts and osteosarcoma [17], implying that IACS-010759 may shift cells from a tumorigenic and undifferentiated profile towards normal osteoblast differentiation. Furthermore, IACS-010759 upregulates the expression of ribosomal protein genes such as *RSP27*, *RPL11*, *RPL32*, *RPL35A*, and *RPL11*, mutations which are found in osteosarcoma-prone Diamond-Blackfan anemia (DBA) patients [37–39]. Homeostasis of ribosomal biogenesis may be an essential process to maintain survival of RTS osteoblasts.

In summary, we established an RTS iPSC disease platform to dissect the pathological mechanisms involved in RTS-associated osteosarcoma (Fig 5J). Transcriptome analysis revealed upregulation of genes involved in OXPHOS, especially mitochondrial respiratory complex I, in RTS osteoblasts. The upregulation of oxidative metabolism in RTS is a druggable pathway, and our preclinical studies using IACS-010759 suggest favorable, clinically relevant changes in RTS osteoblasts treated with this drug. Overall, our study suggests strong clinical potential for therapies targeting mitochondrial respiratory complex I in RTS patients with osteosarcoma.

## Materials and methods

### Ethics statement

The protocol entitled "Molecular Basis of Clinical Spectrum of Rothmund-Thomson Syndrome" (IRB# H-9106) was approved by the Institutional Review Board of Baylor College of Medicine and Affiliated Hospitals. Written formal consent for participation in this IRB-approved protocol was obtained by Dr. Lisa L. Wang (P.I.) from the legally authorized representative of each subject at the time of enrollment.

## Cell culture

RTS and family control (FC) fibroblasts were cultured and maintained in Dulbecco's Modified Eagle Medium (DMEM) supplemented with 10% (vol/vol) fetal bovine serum (FBS) (Opti-Gold performance enhanced F0900-050, GenDEPOT, Barker, TX), L-glutamine, non-essential amino acids, β-mercaptoethanol, and penicillin/streptomycin. iPSC-derived MSCs were maintained in gelatin-coated plates and cultured in DMEM supplemented with 10% FBS, L-glutamine, and penicillin/streptomycin. iPSC-derived osteoblasts were cultured in osteogenic differentiation medium (α-MEM supplemented with 10% FBS, 0.1 μM dexamethasone, 10 mM β-glycerol phosphate, and 200 μM ascorbic acid) [17,40].

## Somatic cell reprogramming RTS and FC fibroblasts

RTS individuals and their parents provided informed consent to participate in an institutional review board-approved study at Baylor College of Medicine (Houston, TX). Sequence analysis of the *RECQL4* gene was performed by Baylor Genetics laboratories (Houston, TX) and confirmed the pathogenic variants in subjects used in this study. Fibroblasts were isolated from skin biopsies from two RTS patients and two first-degree relatives and reprogrammed to iPSCs by transducing Sendai viruses (SeV) expressing the four reprogramming factors OCT4, SOX2, KLF4, and MYC (CytoTune-iPS 2.0 Sendai Reprogramming Kit, Thermo Fisher Scientific, A16517) according to the manufacturer's protocol. The reprogrammed cells were maintained in hESC media (KnockOut DMEM/F12 (Thermo Fisher Scientific, 12660012) containing 20% (vol/vol) KnockOut Serum Replacement (Thermo Fisher Scientific), L-glutamine, non-essential amino acids, β-mercaptoethanol, penicillin/streptomycin, and bFGF). At 3–4 weeks post-induction, individual clones with hESC/iPSC morphology and positive TRA-1-81 staining were selected, passaged on irradiated mouse embryonic fibroblasts (MEFs), and maintained for 10–15 passages. Clones were examined for loss of SeV by genomic DNA PCR using specific primers targeting SeV (5'-GGATCACTAGGTGATATCGAGC-3' and 5'-ATGCACCGCTACGACGTGAGCGC-3'), SOX2 (5'-ATGCACCGCTACGACGTGAGCGC-3' and 5'-AATGTATCGAAGGTGCTCAA-3'), and KLF4 (5'-TTCCTGCATGCCAGAGGAGCCC-3' and 5'-AATGTATCGAAGGTGCTCAA-3').

## Teratoma formation assay

Teratoma formation assay was performed as described previously [41]. H&E staining for histology was performed by Histowiz (Brooklyn, NY).

## *In vitro* differentiation of iPSCs to MSCs

*In vitro* differentiation of iPSCs from RTS individuals and unaffected family members to MSCs was performed by a defined optimized MSC differentiation protocol described previously [27]. Briefly, 80% confluent iPSCs were cultured in StemMACS iPSC-Brew XF medium (Miltenyi Biotec, 130-104-368) supplemented with 10 μM TGFβ inhibitor SB-431542 (MilliporeSigma, S4317) in Matrigel-coated plates at 37˚C and 7.5% $CO_2$ and passaged at 80–90% confluence by Accutase cell detachment solution (StemCell Technology, 07922). After 25 days in culture, cells at the edge of differentiated cell clusters became spindle-shaped. These differentiated cells were trypsinized into single cells and maintained in gelatin-coated plates using modified human ESC-MSC medium (KnockOut DMEM/F-12 supplemented with 10% KnockOut Serum Replacement, L-glutamine, non-essential amino acids, β-mercaptoethanol, 20 ng/ml bFGF2, 10ng/ml EGF, 10 μM SB-431542, and penicillin/streptomycin). The medium was changed daily. When cells reached 80–90% confluence, cells were passaged at a 1 to 3 ratio

for three weeks. The MSCs were characterized by staining for MSC surface markers CD44, CD73, and CD105.

## Osteogenic differentiation and ARS staining

Osteogenic differentiation was carried out based on previously described protocols [17,18,40]. MSCs were seeded at a density of $3 \times 10^5$ cells/per well in a 6-well dish and cultured in osteoblast differentiation medium (αMEM supplemented with 10mM β glycerol phosphate, 2% penicillin/streptomycin, 200μM ascorbic acid, and 10% FBS). The medium was changed every three days. Differentiated osteoblasts were fixed in ice-cold 70% ethanol at room temperature for 2 h, washed with tap water, and stained by ARS solution (0.02g/ml alizarin red S, pH 4.1–4.3) for 30 min. Stained osteoblasts were imaged under a Leica DMi8 microscope.

## qRT-PCR

qRT-PCR was performed as described previously [41]. PCR primers used to detect gene expression included *NANOG* (5'-TTTGTGGGCCTGAAGAAAACT-3' and 5'-AGGGCTGTCCTGAATAAGCAG-3'), *OCT4* (5'-AACCTGGAGTTTGTGCCAGGGTTT-3' and 5'-TGAACTTCACCTTCCCTCCAACCA-3'), *SOX2* (5'-AGAAGAGGAGAGAGAAAGAAAGGGAGAGA-3' and 5'-GAGAGAGGCAAACTGGAATCAGGATCAAA-3'), *COL1A1* (5'-GAGGGCCAAGACGAAGACATC-3' and 5'-CAGATCACGTCATCGCACAAC-3'), *IGF2* (5'-GTGGCATCGTTGAGGAGTG-3' and 5'-CACGTCCCTCTCGGACTTG-3'), *CLEC3B* (5'-CCCAGACGAAGACCTTCCAC-3' and 5'-CGCAGGTACTCATACAGGGC-3'), *NDUFA7* (5'-TGCAGCTACGCTACCAGGA-3' and 5'-GAGCTTGTGGCTAGGACCC-3'), *NDUFB1* (5'- GTCCCTATGGGATTTGTCATTGG-3' and 5'- CAGTTAGCCGTTCATCACTCTT-3'), *NDUFB2* (5'-GGAGGCCGCCTTTTCAGAA-3' and 5'-GGAAGGATCAGGATACGGAAAGT-3'), *NDUFS8* (5'-CCATCAACTACCCGTTCGAGA-3' and 5'-CCGCAGTAGATGCACTTGG-3'), and internal control *GAPDH* (5'-CCACTCCTCCACCTTTGAC-3' and 5'-ACCCTGTTGCTGTAGCCA-3').

## *In vitro* AIG assay

*In vitro* AIG assay was described previously [17]. Briefly, RTS, FC, and WT iPSC-derived MSCs were differentiated into osteoblasts. $1 \times 10^4$ differentiating osteoblasts (D7) were resuspended in osteoblast differentiation medium with 0.4%–0.5% LMP agarose. The cell suspensions were then plated in 12-well plates containing solidified 0.8% agarose in osteoblast differentiation medium. Osteoblasts were maintained in osteoblast differentiation medium for 1.5 months with medium changes every 3 days. Colonies (considered to have a diameter $\geq 50 \mu m$) were counted under a Leica DMi8 microscope.

## Immunofluorescence staining and immunoblotting

Immunofluorescence staining and immunoblotting was described previously [41–43]. Antibodies used in this study included NANOG (R&D Systems, AF1997), OCT4 (Santa Cruz, SC-9081), TRA-1-81-Alexa Fluor 555-conjugated (R&D Systems, FAB3195A), SSEA4-PE conjugated (R&D Systems, FA1435P-025), CD44 (BD PharMingen, 559942), CD73 (BD PharMingen, 550257), CD105 (eBioscience, 12–1057), RECQL4 (MilliporeSigma, SAB1410002), NDUFA7 (ABclonal Science, A8441), NDUFS8 ABclonal Science, A13034), STK39 (ABclonal Science, A2275), EREG (ABclonal Science, A16372), ERK1/2 (Cell Signaling, 4695), phospho-ERK1/2 (Cell Signaling, 9101), and ACTIN (MilliporeSigma, A2228) antibodies.

## RNA-seq, GSEA, and enrichment map analysis

iPSC-derived MSCs and osteoblasts were lysed and RNA samples were prepared by Trizol RNA isolation reagent (Thermo Fisher Scientific, 15596026). RNA-sequencing (RNA-seq) was performed by BGI Genomics (Cambridge, MA) and UTHealth Cancer Genomics Center (Houston, TX). All RNA-seq data analyses were performed using Galaxy Community Hub to calculate FPKM (Fragments Per Kilobase of transcript per Million) as described previously [17]. Differential gene expression was analyzed by DESeq2 [44]. Gene Set Enrichment Analysis (GSEA) was employed to analyze differentially regulated Gene Ontology biological processes (GO_BPs), Kyoto Encyclopedia of Gene and Genomes (KEGG) pathways, Reactome, onco-genic signatures, and transcription factor targets (TFTs) [45]. Differences in gene sets were considered significant if p-value <0.05 and FDR q-value <0.25. Enrichment map analysis was performed to analyze enriched GO_BPs in osteosarcoma tissues using a published osteosar-coma microarray dataset (GSE36001) [46].

## Cell proliferation assay

RTS and FC osteoblasts were seeded in flat bottom 96-well plates (Corning, 3598) at a density of 3,000 cells per well and cultured in 200 µl osteoblast differentiation medium with or without IACS-010759 (dissolved in DMSO) for 3 days. Cell viability upon IACS-010759 treatment was examined by PrestoBlue cell viability reagent (Thermo Fisher Scientific, A13262) following the manufacturer's instructions. The quantitative measure of viability was read by absorbance at 570 nm.

## Sequencing data processing and variant calling

WES of RTS fibroblasts and RTS-B2 and RTS-B3 iPSCs was performed using Novogene exome hybridization protocol followed by barcoding and sequencing of paired-end 150-bp reads on an Illumina NovaSeq 6000 instrument (Illumina Inc., San Diego, CA). Trim Galore! (Version 0.6.6) was then applied for adapter and quality trimming to the FastQ files (DOI: https://doi.org/10.14806/ej.17.1.200). Sequencing reads mapping to the human genome (GRCh38/hg38), initial variant calling and quality filtration were performed using the best practices in BWA-GATK4 (v4.2.2.0) pipeline with the tumor-only mode on [47,48]. Single nucleotide variants were detected using MuTect2 [49], while indels were not interrogated given the high rates of false-positive calls. Additional filters were applied to eliminate false-pos-itive somatic variant calls: 1) read depth lower than 25th percentile of the sequencing coverage [50], 2) read depth higher than 75th percentile of the sequencing coverage [51], 3) present in dbSNP [52]. Functional annotation of the remaining somatic variants was carried out using ANNOVAR (v0.4.0.0) [53].

## Cell cycle analysis

G0/G1, G2, and S-phase cells were stained with PI, examined by the BD flow cytometer, and analyzed by FlowJo as previously described [54].

## Reactive oxygen species assay

Intracellular oxidation of osteoblasts was assayed by measuring the conversion of DCFH to DCF using the Reactive Oxygen Species (ROS) Fluorometric Assay Kit (Elabscience, E-BC-K138-F) according to the manufacturer's instructions. In short, DCFH-DA was added to osteoblasts and incubated for 1 h, then fluorescence of DCF was measured at 525nm using TECAN Infinite M Plex multimode microplate reader.

### Senescence staining

Senescent cells were measured using the Senescence β-Galactosidase Staining Kit (Cell Signaling Technology, #9860). Osteoblasts, either treated with 100 nm IACS-010759 or DMSO for 3 days, were washed, fixed, then stained with β-Galactosidase overnight at 37˚C. The population of β-Galactosidase positive cells was calculated by ImageJ.

### Apoptotic activity assay

Apoptosis was measured via Caspase 3/7 activity using the SensoLyte Homogeneous AMC Caspase-3/7 Assay Kit (Anaspec, AS-71118). The assay was carried out according to the manufacturer's instructions. FC and RTS osteoblasts were cultured in a 6-well plate and then treated with 100 nm IACS-010759 for 3 days. Cell lysates were incubated with Caspase 3/7's fluorogenic substrate Ac- Asp-Glu-Val-Asp (DEVD) -AMC for 4 h in the dark. Fluorescence was measured at Ex/Em = 354nm/442nm using TECAN Infinite M Plex multimode microplate reader.

### ATP detection assay

ATP levels of RTS and FC Day 24 osteoblasts were measured using the ATPlite Luminescence Assay System (Perkin Elmer, 6016943) and following the manufacturer's instructions. Briefly, cell lysis was incubated with ATPlite solution containing luciferase and D-Luciferin firefly. The ATP concentration is proportional to the emitted luminescence. The samples were incubated at room temperature in the dark for 10 minutes. Luminescence was measured by TECAN Infinite M Plex multimode microplate reader. Results were compared to the supplied ATP standard.

### Electron transport chain enzymatic assays

Electron transport chain (ETC) enzymatic activity and citrate synthase (CS) activity assays were performed as previously described [55,56]. The total protein content of osteoblast lysates was quantified by Bradford protein assay (Bio-Rad, 5000006) and used for normalization.

### Seahorse assay

Seahorse assays (Agilent) were carried out according to the manufacturer's protocol with modifications as described below. 30,000 osteoblasts were seeded into each well of a Seahorse cell culture plate, with at least 4 wells per assay left blank as controls. Osteoblasts were cultured in 200 µl osteoblast differentiation medium at 37˚C for 3 days. A Seahorse assay plate was prepared by incubating in water overnight and replacing with calibration medium for 4 h prior to performing the assay. Each well in the Seahorse assay plate was filled with 180 µl of assay medium supplemented with 10 mM pyruvate, 2 mM glutamine, 25 mM glucose, and 4 mg/ml BSA. The tissue culture plate was then incubated at 37˚C with 0% $CO_2$ and 0% humidity. Assay chemicals were dissolved in the assay medium without BSA.

Assay measurements were made at baseline and every 6 min over 18 min (3 cycles each entailing a 3-min mix and 3-min measurement). Next, IACS 010759 was injected from port A over 30 min and measurements were made at time 0 and at 6-min intervals over 30 min (5 cycles each of 3-min mix and 3-min measurement). Lastly, 15 µM Oligomycin, 30 µM FCCP, and 5 µM Rotenone/Antimycin A were injected for 18 min from port B. Immediately following each assay, osteoblasts were fixed with 4% paraformaldehyde and stained with DAPI (1:5000 in DPBS). Cell numbers were counted using a Biotek Lionheart automatic microscope with default settings for TPP 96 well TC flat bottom plates. Nuclei were counted by

thresholding for a minimum size of 5 μm and a maximum size of 50 μm including edge objects. The rates of change of dissolved oxygen (OCAR) and pH (ECAR) were calculated using the Wave software, normalized to total cell number.

## Statistical analysis

Statistical analysis was performed in Prism 8.0 with a two-tailed Student's t-test. Results are expressed as the mean ± SEM. Difference between two groups was compared by two-tailed unpaired Student's t-test or ANOVA in Prism 8.0. ns, non-significant; *, $p < 0.05$; **, $p < 0.01$; and ***, $p < 0.001$.

## Supporting information

**S1 Fig. Characterization of RTS and FC iPSCs.** (A) *In vivo* teratoma formation assay demonstrates three germ-layer differentiation abilities of RTS and FC iPSCs. Scale bar, 200 μm. (B) PCR detection of SeV genome and transgenes indicates that RTS and FC iPSCs are footprint-free. (C) Somatic variant analysis of RTS-B2 and RTS-B3 iPSCs suggests that limited somatic mutations are generated during cellular reprogramming.
(TIF)

**S2 Fig. Characterization of RTS and FC iPSCs, MSCs, and osteoblasts.** (A) Relative cell proliferation of RTS and FC iPSCs (upper panel), MSCs (middle panel), and osteoblasts (lower panel). n = 3 biological replicates. (B) Cell cycle profiles of RTS and FC iPSCs, MSCs, and osteoblasts are performed using PI staining by flow cytometry. (Upper) Cell cycle profiles of RTS iPSCs, MSCs, and osteoblasts. (Lower) Bar histogram shows the percentage of cells in the sub-G1, G0/G1, S, and G2/M phases. n = 3 biological replicates. (C) Wound healing assay determines the cell migration ability of RTS and FC osteoblasts. Wound healing assays were performed at 72 hours in RTS and FC osteoblasts. (Left) Representative phase-contrast microscope images show the area covered by RTS and FC osteoblasts at 72 hours after wounding. (Right) The cell migration rate is determined by the rate of cells moving towards the scratched area upon time measured. n = 3 biological replicates.
(TIF)

**S3 Fig. Transcriptional analysis of RTS osteoblasts and osteosarcoma.** (A) Reactome analysis demonstrates enrichment of multiple canonical pathways in RTS compared to FC MSCs (D0), pre-osteoblasts (D15), and osteoblasts (D24). (B) Left, GO_BPs enriched in osteosarcoma by EnrichmentMap analysis. Network visualization of gene sets enriched in osteosarcoma compared with normal bone tissues and osteoblasts (p value <0.05, FDR q value <0.1) indicates that GO_BPs involved in DNA repair, chromatin regulation, cell cycle, and mitochondrial ATP production and repair are enriched in osteosarcoma. The number of enriched genes in each GO is displayed as node size, with closer distance between nodes representing increased overlap between genes in ontologies. Enriched gene sets in osteosarcoma are displayed in red and enriched gene sets in bone tissues and osteoblasts in blue. Right, representative enriched gene sets (ATP synthesis coupled electron transport and respiration electron transport chain) involved in mitochondrial ATP production are shown.
(TIF)

**S4 Fig. Metabolic differences between RTS and FC osteoblasts.** (A-E) Enzyme activity of complexes I and III (A), complex II (B), complexes II and III (C), complex IV (D), and citrate synthase (E) show no significant difference between RTS and FC osteoblasts, leaving complex I as the only component of the electron transport chain with upregulated activity. n = 3 biological replicates. (F) The decrease of total ATP levels is observed in RTS osteoblasts compared to

FC osteoblasts. n = 4 biological replicates.
(TIF)

**S5 Fig. Biological effects and transcriptome alterations in IACS-010759-treated RTS osteoblasts.** (A) IACS-010759 alters RTS osteoblast cell morphology from a spindle shape to an irregular shape. Scale bar, 100 μm. (B) Cell proliferation assay indicates that Rotenone impairs cell proliferation of RTS and FC osteoblasts. n = 4 biological replicates. (C) ROS activity assay demonstrates that IACS-010759 does not induce ROS production in both RTS and FC osteoblasts. n = 3 biological replicates. (D) Cell apoptosis assay indicates a decrease of Caspase 3/7 activities upon IACS-010759 treatment in both RTS and FC osteoblasts. n = 3 biological replicates. (E) Cell cycle profiles of RTS and FC osteoblasts are performed upon IACS-010759 treatment. (Upper) Cell cycle profiles of IACS-010759-treated RTS and FC osteoblasts. (Lower) Bar histogram shows the percentage of cells in the sub-G1, G0/G1, S, and G2/M phases. n = 3 biological replicates. (F) Representative GO_BPs influenced by IACS-010759, including protein targeting to membrane, translational initiation, positive regulation of MAPK activity, and positive regulation of cell cycle process. (G) KEGG pathway analysis confirms upregulation of ribosome function in IACS-010759-treated RTS osteoblasts.
(TIF)

**S1 Table. The mutation analysis of RTS-B2 and RTS-B3 iPSCs.**
(XLSX)

## Acknowledgments

The authors gratefully acknowledge John F. Hancock, Jeffrey T. Chang, Jeffrey A. Frost, Kartik Venkatachalam, and Guang Peng for suggestions and reagents; Ta-Tara Rideau at Baylor College of Medicine for clinical research and data management support; and UTHealth Cancer Genomics Core for technical support. We also thank the Translational Research to AdvanCe Therapeutics and Innovation in ONcology (TRACTION) and Institute for Applied Cancer Science (IACS) Platforms at The University of Texas MD Anderson Cancer Center for providing IACS-010759 under an MTA. Finally, we thank the patients and families who participated in the research to make this study possible.

## Author Contributions

**Conceptualization:** Brittany E. Jewell, Lisa L. Wang, Dung-Fang Lee.

**Formal analysis:** Brittany E. Jewell, An Xu, Dandan Zhu, Mo-Fan Huang, Linchao Lu, Jun Hyoung Park, Benny Abraham Kaipparettu, Lisa L. Wang, Dung-Fang Lee.

**Funding acquisition:** Holger K. Eltzschig, Zhongming Zhao, Benny Abraham Kaipparettu, Ruiying Zhao, Lisa L. Wang, Dung-Fang Lee.

**Investigation:** Brittany E. Jewell, An Xu, Dandan Zhu, Mo-Fan Huang, Linchao Lu, Mo Liu, Erica L. Underwood, Jun Hyoung Park, Huihui Fan, Julian A. Gingold, Ruoji Zhou, Jian Tu, Zijun Huo, Ying Liu, Weidong Jin, Yi-Hung Chen, Yitian Xu, Shu-Hsia Chen, Nino Rainusso, Nathaniel K. Berg, Danielle A. Bazer, Holger K. Eltzschig, Zhongming Zhao, Benny Abraham Kaipparettu, Ruiying Zhao.

**Resources:** Christopher Vellano, Philip Jones.

**Supervision:** Ruiying Zhao, Lisa L. Wang, Dung-Fang Lee.

**Validation:** Lisa L. Wang, Dung-Fang Lee.

**Writing – original draft:** Brittany E. Jewell, Julian A. Gingold, Lisa L. Wang, Dung-Fang Lee.

**Writing – review & editing:** Lisa L. Wang, Dung-Fang Lee.

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
