## [Decision Letter · Decision Letter 0]

22 Jul 2021

Dear Dr Lee,

Thank you very much for submitting your Research Article entitled 'Patient-derived iPSCs Link Elevated Mitochondrial Respiratory Complex I Function to Osteosarcoma in Rothmund-Thomson Syndrome' to PLOS Genetics.

The manuscript was fully evaluated at the editorial level and by independent peer reviewers. The reviewers appreciated the attention to an important problem, but raised some substantial concerns about the current manuscript. Based on the reviews, we will not be able to accept this version of the manuscript, but we would be willing to review a much-revised version. We cannot, of course, promise publication at that time.

If you decide to revise the manuscript for further consideration at PLOS Genetics, please aim to resubmit within the next 60 days, unless it will take extra time to address the concerns of the reviewers, in which case we would appreciate an expected resubmission date by email to plosgenetics@plos.org.

[LINK]

We are sorry that we cannot be more positive about your manuscript at this stage. Please do not hesitate to contact us if you have any concerns or questions.

Yours sincerely,

Joseph Opferman

Associate Editor

PLOS Genetics

Peter McKinnon

Section Editor: Cancer Genetics

PLOS Genetics

Reviewer's Responses to Questions

**Comments to the Authors:**

Reviewer #1: The manuscript from Jewell et al describes the derivation and analysis of iPSC derived from two patients with Rothmund-Thomson Syndrome. Using this approach they generated iPSC and used them to attempt to understand the role of RECQL4 mutations in the osteoblastic lineage through directed differentiation. Using this approach with transcriptome analysis, the authors identified that the patient derived cells had elevated mitochondrial respiratory complex I function. They validate this using the seahorse method and identify that the inhibitor IACS-010759 was relatively selectively toxic to the RTS patient derived osteoblasts compared to parental derived lines.

Overall this is an interesting and thought provoking study. I think it will be valuable to the research community and may be a means to more completely understand the mechanisms through which biallelic mutations in RECQL4 cause disease mainfestations and cancer.

Specific comments:

1. The authors do not provide any evidence that the RECQL4 mutations are present in the iPSC cells derived from the patients - whilst it is assumed (and I think this assumption is reasonable) the mutations should be sequenced and provided from the iPSC. The authors also should provide an assessment, and if possible a western, to understand if the mutation generates a predicted protein product and if so how and where does it localise. Ideally using the endogenous protein but otherwise with mutations in a cDNA.

2. Do you know the mutational burden of the RTS patient derived cells to be able to conclude that the observed phenotypes are solely due to RECQL4 mutations? Exome sequencing would resolve this.

3. One of the major advantages of patient derived iPSC is the ability to use human cells directly to understand the biology. It also offers the powerful experimental model of generating gene corrected isogenic cell lines that can be used to allow the most rigorous and robust comparison possible. It is not clear to me why the authors have not attempted to use RECQL4 corrected isogenic cell lines to undertake the analysis in place of, or at least alongside, those derived from a heterozygous parent. The analysis of isogenic pairs of cells would potentially discern subtle biology and also control for all other mutations that may be present in any individuals sample. Whilst the parent derived cells are a control, they are derived from a significantly different age of human, have had different exposures and responses to exposures for life.

4. Cells proliferating in soft agar are a poor substitute for in vivo tumor assays (text associated with this observation could be tempered in my opinion). I would encourage the authors to either take the osteoblasts and transplant in nude mice or NSG in matrigel and demonstrate that they are tumorigenic.

5. page 10; lines 210-213 - the text states that sporadic osteosacroma cell lines have increased ATP production pathways as well. Does this imply that it is a feature of osteosarcoma per se rather than reflective of RECQL4 biology? It is known that loss or mutations in p53, Terc and Rb cause metabolic reprogramming in cells and mice potentially pointing to a more generic feature of tumor suppressors. This is more a comment that a request to do any experiments but may be worth considering.

6. Previous work has assessed energy metabolism in murine Recql4 mutant bearing cells (PMID: 33361189). This publication did not report and difference in metabolic activity. Do you think this reflects cell type differences, species differences or something else. Given the broad spectrum of clinical manifestations and that these affect cell lineages derived from different germ layers can the authors speculate as to why there is only a discernable difference following osteoblastic differentiation and not one that is apparent in the iPSC or MSC?

7. At a conceptual level, can the authors discuss further if they think the inhibitor IACS-010759 would be applicable for treating osteosarcoma in patients with germ-line mutations in RECQL4 (such as the RTS patients) where multiple cell types in the body may be affected by exposure to this agent as all cells have the mutation in RECQL4?

8. Demonstrate that the change in MAPK pathway (Fig 5f) is reflected at the protein level and not just from transcriptome signatures.

9. Provide additional baseline characteristics of the populations and cells, for example proliferation rates of iPSC, MSC and osteoblasts, cell cycle profiles of the osteobalstic populations and following treatment, evidence of any DNA damage response being active in the cells from MSC to osteogenic cells.

Reviewer #2: Patient-derived iPSCs Link Elevated Mitochondrial Respiratory Complex I Function to

Osteosarcoma in Rothmund-Thomson Syndrome by Jewell et al.

In this manuscript the authors have attempted to dissect the molecular mechanism behind the development of osteosarcoma in Rothmund-Thomson syndrome patients. Using patient derived induced pluripotent stem cells (iPSC) they demonstrated defective osteogenic differentiation. Further transcriptomic analysis showed increased mitochondrial complex I gene expression. Finally, authors used a drug, IACS-010759 to inhibit complex I activity to show potential intervention. Overall, the study is o interest, as they report establishment of IPSCs for RTS. However, the studies performed on these cells are not very extensive, and it as already well established that RECQL4 plays a role in mitochondria. Additional explanations and experiments are needed

Concerns:

1. Figure 3F: its not sufficient to show the oncogenic potential just by colony formation assay. Authors should try other approach like in vitro metastatic assay to show the oncogenic property of patient derived osteoblasts.

2. Figure 4 A: showed increase in complex I activity. It is important to show the complex I protein level in RTS patients. It would be interesting to check how OXPHOS assembly is affected in RTS patients cell lines.

3. Figure 4 E: Decreased ATP levels in RTS patient fibroblasts has been reported previously (Kumari et al, 2106). Hence authors should check the total ATP levels in RTS osteoblasts.

4. Figure 5 C: complex I inhibition by IACS-010759 treatment increased cell viability in control cells but decreased in RTS patient cells. Please explain.

5. Figure 5: rotenone, a known inhibitor for complex I should be used in cell viability assay in parallel with IACS-010759. This will tell us whether IACS-010759 is a better candidate.

6. Figure 5 C: decrease in cell viability should be supported by analyzing other parameters like senescence and apoptosis.

7. RTS patient cell lines exhibit increased reactive oxygen species (ROS) production. Authors should check the ROS levels after IACS-010759 treatment.

8. Connections need to be discussed relating to the function of RECQL4 in mitochondria, where a number of studies have been done

Reviewer #3: The study by Jewell et al., uses patient-derived iPSCs to study mechanisms involved in RECQL4 mutation-induced osteosarcoma progression of patients with Rothmund-Thomson Syndrome (RTS). The authors, derived iPSCs from four individuals in two families with RTS (one unaffected and one affected in each family). Affected members analyzed developed osteosarcoma. The pluripotency and differentiation of iPSC lines into all three germ lines were carefully examined. They found that osteoblastic differentiation was compromised in RTS as compared to control lines and RTS osteoblasts exhibited increased clonal growth ability in soft agar. Further, three stages of osteogenic differentiation were collected for RNA-Seq analyses which revealed an enrichment of mitochondrial energy production in RTS osteoblasts relative to controls. They specifically identified an increase in mitochondrial respiratory complex I genes in RTS osteoblasts. Functional assays further support these initial results. Finally, they show that OXPHOS and transcripts related to cell growth are reduced in RTS osteoblasts treated with IACS-010759, a mitochondrial complex I inhibitor supporting the notion that mitochondrial complex I is implicated in RTS osteosarcoma progression. The authors have performed enormous amount of work, and established a useful model for the analyses of RTS and its progression to osteosarcoma. Their data is solid and conclusions overall well supported by the data. My main comment is regarding Figure 5.

Figure 5C, seems to be a viability and not cell growth measurement. This needs to be clarified, and cell growth plots provided.

In addition, it would be important for the authors to confirm reduced protein expression of some of the transcripts they described to be reduced in response to the Complex I inhibitor.

**Have all data underlying the figures and results presented in the manuscript been provided?**

Reviewer #1: Yes

Reviewer #2: Yes

Reviewer #3: Yes

PLOS authors have the option to publish the peer review history of their article (what does this mean?). If published, this will include your full peer review and any attached files.

Reviewer #1: No

Reviewer #2: No

Reviewer #3: No

---

## [Decision Letter · Decision Letter 1]

29 Nov 2021

Dear Dr Lee,

We are pleased to inform you that your manuscript entitled "Patient-derived iPSCs Link Elevated Mitochondrial Respiratory Complex I Function to Osteosarcoma in Rothmund-Thomson Syndrome" has been editorially accepted for publication in PLOS Genetics. Congratulations!

Yours sincerely,

Joseph Opferman

Associate Editor

PLOS Genetics

Peter McKinnon

Section Editor: Cancer Genetics

PLOS Genetics

Comments from the reviewers (if applicable):

Reviewer's Responses to Questions

**Comments to the Authors:**

Reviewer #1: The authors have addressed the majority of concerns raised during review.

The following are comments for the authors:

1. I dont think "in vitro tumorigenic ability" is a concept of merit when the in vivo data with the same cells doesn't demonstrate tumor formation, but this is a subjective view. I think it is better characterised as altered in vitro proliferation/behavior. I would suggest moderating of this text around this concept but I dont think this is absolutely necessary.

2. I accept there are difficulties with gene correction but the reasons supplied are just sufficient to address this comment. In the interests of time and not making unreasonable reviewer requests I accept the authors statements. Prime editing and single base correction is now relatively accessible.

3. I do find it striking that no assessment of actual RTS osteosarcomas was considered or provided (eg RNA-seq and demonstrating pathway overlaps; primary cells isolated from the osteosarcoma for analysis) as this would potentially provided very strong support for the authors findings.

Reviewer #2: They have improved the paper and largely responded well to the issues I had, so I am in favor of publication

Reviewer #3: The authors have responded carefully to all comments.

**Have all data underlying the figures and results presented in the manuscript been provided?**

Reviewer #1: Yes

Reviewer #2: Yes

Reviewer #3: Yes

PLOS authors have the option to publish the peer review history of their article (what does this mean?). If published, this will include your full peer review and any attached files.

Reviewer #1: **Yes: **Carl Walkley

Reviewer #2: No

Reviewer #3: No

**Data Deposition**

http://datadryad.org/submit?journalID=pgenetics&manu=PGENETICS-D-21-00811R1

**Press Queries**

---

## [Editor Report · Acceptance letter]

10 Dec 2021

PGENETICS-D-21-00811R1 

Patient-derived iPSCs Link Elevated Mitochondrial Respiratory Complex I Function to Osteosarcoma in Rothmund-Thomson Syndrome 

Dear Dr Lee, 

We are pleased to inform you that your manuscript entitled "Patient-derived iPSCs Link Elevated Mitochondrial Respiratory Complex I Function to Osteosarcoma in Rothmund-Thomson Syndrome" has been formally accepted for publication in PLOS Genetics! Your manuscript is now with our production department and you will be notified of the publication date in due course.

With kind regards,

Zsofia Freund

PLOS Genetics

On behalf of:
